# Multi-targeting of K-Ras domains and mutations by peptide and small molecule inhibitors

**Mansour Poorebrahim**[1], **Mohammad Foad Abazari**[2], **Leila Moradi**[3], **Behzad Shahbazi**[3], **Reza Mahmoudi**[4], **Hourieh Kalhor**[5], **Hassan Askari**[6], **Ladan Teimoori-Toolabi**[3]*

1 Targeted Tumor Vaccines Group, Clinical Cooperation Unit Applied Tumor Immunity, German Cancer Research Center (DKFZ), Heidelberg, Germany, 2 Research Center for Clinical Virology, Tehran University of Medical Sciences, Tehran, Iran, 3 Molecular Medicine Department, Biotechnology Research Center, Pasteur Institute of Iran, Tehran, Iran, 4 Department of Medical Biotechnology, Faculty of Advanced Medical Sciences, Tabriz University of Medical Sciences, Tabriz, Iran, 5 Cellular and Molecular Research Center, Qom University of Medical Sciences, Qom, Iran, 6 Gastroenterohepatology Research Center, Shiraz University of Medical Sciences, Shiraz, Iran

* lteimoori@pasteur.ac.ir

**Data Availability Statement:** The data that support the findings of this study are available in the manuscript and its Supporting information files.

## Abstract

K-Ras activating mutations are significantly associated with tumor progression and aggressive metastatic behavior in various human cancers including pancreatic cancer. So far, despite a large number of concerted efforts, targeting of mutant-type K-Ras has not been successful. In this regard, we aimed to target this oncogene by a combinational approach consisting of small peptide and small molecule inhibitors. Based on a comprehensive analysis of structural and physicochemical properties of predominantly K-Ras mutants, an anti-cancer peptide library and a small molecule library were screened to simultaneously target oncogenic mutations and functional domains of mutant-type K-Ras located in the P-loop, switch I, and switch II regions. The selected peptide and small molecule showed notable binding affinities to their corresponding binding sites, and hindered the growth of tumor cells carrying K-Ras$^{G12D}$ and K-Ras$^{G12C}$ mutations. Of note, the expression of K-Ras downstream genes (i.e., CTNNB1, CCND1) was diminished in the treated Kras-positive cells. In conclusion, our combinational platform signifies a new potential for blockade of oncogenic K-Ras and thereby prevention of tumor progression and metastasis. However, further validations are still required regarding the *in vitro* and *in vivo* efficacy and safety of this approach.

## Author summary

K-Ras activating mutations are associated with tumor progression and aggressive metastatic behavior in cancers. We aimed to target this mutated protein as an oncogene with small peptides and small molecules. The selected peptide and small molecules by computational methods showed notable binding affinities to mutated and oncogenic K-Ras. Also, they hindered the proliferation of pancreatic tumor cells. These compounds

**Funding:** This study was funded by Pasteur Institute of Iran (Grant Number: 583). This fund was given to L.T.T. M.P. is supported by Alexander von Humboldt Foundation fellowship. The funders had no role in study design, data collection and analysis, decision to publish, or preparation of the manuscript.

**Competing interests:** The authors have declared that no competing interests exist.

diminished the expression of downstream genes to mutant K-Ras too. Our combinatorial approach introduces new candidates for blockade of oncogenic K-Ras which is observed in many types of cancer. The effect of these compounds should be validated by further in vitro and in vivo analysis.

## 1. Introduction

Members of the Ras (rat sarcoma) family are a group of GTP-binding proteins with low molecular weight (~21 kDa) that act as molecular switches regulating a multitude of cellular processes, such as cell differentiation, proliferation, and survival. There are three main members of this family including, H-Ras, N-Ras, and K-Ras (splice variants K-Ras4A and K-Ras4B) which play a role in cancer initiation and progression. Ras family oncogenes are mutated in approximately 30% of human cancers. Mutations in these oncogenes lead to their permanent GTP-bound conformation and, in turn, induce cell motility and uncontrolled cell proliferation through activating downstream signaling pathways (i.e., Wnt/β-catenin, VEGFA, and FGF2 pathways) [1–3]. Among this family, K-Ras is the most frequently mutated isoform in human cancers (about 86% of total), followed by N-Ras (11%) and H-Ras (3%). Mutations in K-Ras have been observed in 90% of pancreatic adenocarcinomas, 45% of colorectal cancers, and 35% of lung adenocarcinomas [4–6].

Activation of K-Ras is accomplished by the cooperation of three distinct domains including P-loop, switch I, switch II, which are important for K-Ras interactions with GTP/GDP and effector proteins. In spite of more than two decades of working on the development of small molecule inhibitors, GTP/GDP-binding site of K-Ras has thus far proven to be undruggable [7]. To date, there is no promising therapeutic strategy to specifically block the GTP/GDP-binding pocket of K-Ras. Indeed, due to the picomolar binding affinity of Kras-GTP and micromolar concentration of GTP in the cell, targeting of GTP-binding site with a competitive inhibitor is not plausible [8]. Furthermore, targeting other K-Ras binding interfaces (i.e., binding sites for Raf, PI3K, RalGDS, and GAP) is not very feasible due to challenges in small molecule screening and identification of the precise protein-protein interaction (PPI) surfaces [9]. Therefore, single targeting of the characterized binding pockets of K-Ras seems to have imperfect results. In this study, for the first time, we aimed to simultaneously target P-loop, switch I, and switch II domains of K-Ras via a combination of peptide and small molecule inhibitors. For decades, the use of bispecific peptides capable of dual targeting biological macromolecules has been investigated in clinical trials and these compounds are known as a promising strategy for the treatment of human cancers [10]. It is reasonable that the function of these peptides can be increased when a synergistic small molecule is combined. As expected, the preliminary results showed that the combinational therapeutic strategy used in this study can be useful for the treatment of patients with K-ras-related cancers.

## 2. Methods and material

### 2.1 Preparation and analysis of the wild-type and mutant-type K-Ras structures

Structures of the wild-type K-Ras (ID: 4LPK) and its six frequent mutants including K-Ras$^{G12C}$ (ID: 4LDJ), K-Ras$^{G12D}$ (ID: 5XCO), K-Ras$^{G12R}$ (ID: 4QL3), K-Ras$^{G12V}$ (ID: 4TQ9), K-Ras$^{G13D}$ (ID: 5UQW) and K-Ras$^{Q61H}$ (ID: 3GFT) were obtained from protein data bank (PDB) (https://www.rcsb.org/pdb/home/home.do). The structures were subsequently

**Table 1. Physicochemical properties and number of assays and cell lines used for validation of anti-cancer capability of studied peptides.**

| Peptide Name | Peptide Length | Molecular weight (g/mol) | pH(I)* | Charge at pH 7 | No. of Assays | No. of Cell lines | Water solubility |
|---|---|---|---|---|---|---|---|
| FLAK93 | 5 aa | 720.9 | 10.69 | +2 | 33 | 117 | Good |
| FLAK50T1 | 10 aa | 1102.41 | 10.98 | +3 | 16 | 96 | Good |
| FLAK50T6 | 11 aa | 1230.58 | 11.15 | +4 | 11 | 66 | Good |
| FLAK94 | 7 aa | 732.91 | 9.88 | +1 | 11 | 65 | Poor |
| C-10 | 13 aa | 1713.16 | 11.28 | +5.1 | 7 | 21 | Good |
| Tat | 9 aa | 1339.61 | 12.81 | +8 | 6 | 6 | Good |
| FLAK50 Z1 | 13 aa | 1344.69 | 11.15 | +4 | 4 | 24 | Good |
| LfcinB | 12 aa | 1753.16 | 11.9 | +5.9 | 4 | 20 | Good |
| Temporin-La | 13 aa | 1624.02 | 10.26 | +2.1 | 2 | 18 | Good |
| LL-37(17–29) | 13 aa | 1719.09 | 11.87 | +4 | 5 | 13 | Good |
| GA-K4 | 11 aa | 1494.86 | 11.15 | +4 | 1 | 9 | Good |
| Cationic Amphiphilic | 11 aa | 1872.33 | 11.16 | +4 | 3 | 6 | Good |
| FLAK-120G | 11 aa | 1167.53 | 10.99 | +3 | 2 | 8 | Good |
| Halictine 1 | 12 aa | 1410.73 | 11.41 | +2.1 | 2 | 4 | Poor |
| (HHPHG)2 | 10 aa | 1149.18 | 7.97 | +0.6 | 3 | 3 | Poor |
| 17 | 16 aa | 1698.19 | 11.16 | +4 | 1 | 9 | Poor |
| CAME-15 | 15 aa | 1771.28 | 11.28 | +5 | 1 | 10 | Good |
| RGD-La | 16 aa | 1952.35 | 10.18 | +2.1 | 1 | 9 | Good |
| Retro | 16 aa | 1615.95 | 9.93 | +1 | 1 | 9 | Poor |

visualized and analyzed using PyMOL and UCSF Chimera programs [11,12]. Detailed analysis of K-Ras interactions with ligands was performed using LigPlot + [13].

## 2.2 Construction of peptide and small molecule libraries

Anti-cancer peptides were screened in CancerPPD database based on the defined criteria (i.e., positive surface potential, the high number of assays and cell lines used for validation of anti-cancer activity) and a library of peptides was constructed for subsequent steps [14]. Indeed, we used the previously validated anti-cancer peptides and optimized their binding affinity to K-Ras for application in Kras-related cancers. Since the relevant target sites of K-Ras had a high negative surface potential, the positive surface potential of peptides was considered as an important criterion. According to the surface electrostatic potential and other defined criteria, the final peptide library consisting of 19 peptides was constructed (Table 1 and S1 Fig). In parallel, the ZINC database was used to construct a library of small molecules screened for their structural similarity with trisodium guanosine 5'-[β,γ-imido] triphosphate (GNP), a non-hydrolyzable analog of GTP that binds and irreversibly activates G proteins [15]. By using the FAF-Drugs4 server, the resultant library was filtered based on absorption, distribution, metabolism, excretion, and toxicity (ADMET) profile to enhance the chance of finding molecules with druggable properties. Finally, the small molecules were further filtered based on quantitative estimate of druglikeness (QED) method using FAF-QED service of FAF-Drugs4. This method calculates a physicochemical score for each compound utilizing eight descriptors. This score is ranged from zero (unfavorable) to one (favorable) [16].

## 2.3 Affinity maturation of peptides

We chose the peptides with high anti-cancer activity and optimized their binding affinity and specificity to the mutant-type K-Ras by affinity maturation approach [17]. In this method, for each selected peptide from the original library, a new peptide library with positional

substitutions was generated, in which each residue of the peptide was substituted with all other amino acids. The new peptide libraries were screened for their binding affinity to K-Ras by docking studies, and important residues involved in the stronger interaction with K-Ras were identified.

## 2.4 Molecular docking studies and molecular dynamics simulations

First, a high-resolution docking was carried out between each library's peptide and K-Ras mutants utilizing the Haddock server [18]. Then, the binding energy of each peptide-protein complex was calculated by the PRODIGY tool and top-scored peptide-Kras complexes were identified [19]. Finally, two flexible docking algorithms including SwarmDock and FlexPep-Dock were used for further confirmation of selected peptides. These servers allow full flexibility to the peptide backbone and to all side chains to generate near-native peptide-protein models with low energy conformations [20,21]. In addition, we used a version of AutoDock Vina program, Smina, to perform docking calculations of binding of small molecules to K-Ras$^{G12D}$. This program is a fork of Vina specially optimized to support minimization and new scoring functions [22]. The binding free energy of molecule-protein complexes was calculated and small molecules with the highest binding affinity to K-Ras$^{G12D}$ were identified. For the purpose of dynamics investigation, Molecular Dynamics (MD) trajectories were generated for both selected peptide-protein and molecule-protein complexes. GROMACS version 5 program was used to simulate several runs up to 50ns according to the GROMOS96 force-field [23,24].

## 2.5 Calculation of physicochemical properties and binding energies

The Innovagen's peptide property calculator software (http://pepcalc.com) was used for the calculation of several physicochemical properties of the selected peptides including molecular weight, net charge at pH 7, isoelectric point (pI). The PRODIGY webserver was used for the calculation of Gibbs free energy changes ($\Delta G$) and equilibrium dissociation constant ($K_d$) of peptide-Kras interactions [19]. The binding energies were calculated as follow:

$$\Delta G_{interaction} = -0.09459\,ICs_{charged/charged} - 0.10007\,ICs_{charged/apolar} + 0.19577\,ICs_{polar/polar}$$
$$- 0.22671\,ICs_{polar/apolar} + 0.18681\,\%NIS_{apolar} + 0.3810\,\%NIS_{charged} - 15.9433$$

Where, $ICs_{xxx/yyy}$ is the number of interfacial contacts which are found between the interface of the first and the second interactors. These contacts are classified based on the polar/apolar/charged nature of the interacting residues (i.e., $ICs_{charged/apolar}$ is the number of ICs between charged and apolar residues). The contacts between two residues were defined if any of their heavy atoms was within a distance of 5.5 Å.

The binding energies of small molecule-Kras complexes were calculated using a standard Smina utility.

## 2.6 Cell lines and cell culture conditions

Two pancreatic cancer cell lines including AsPC-1 and MIA PaCa-2 were used in this study. These cell lines have mutations in the K-Ras (AsPC-1: K-Ras$^{G12D}$, MIA PaCa-2: K-Ras$^{G12C}$) [25]. The cell lines were obtained from National Cell Bank of Institute Pasteur of Iran (Tehran, Iran). The AsPC-1 cells were cultured in RPMI-1640 medium, supplemented with 20% (vol/vol) FBS, 2mM glutamine, 100 unit/mL penicillin, and 100μg/mL streptomycin. MIA PaCa-2 cells were also cultured in the same conditions, but 10% FBS. Cell cultures were incubated at 37˚C in a humidified atmosphere of 5% (vol/vol) $CO_2$.

**Table 2. Sequence of Primers.**

| Primer name | Sequence 5′ to 3′ |
|---|---|
| CCND-1-F | CCGTCCATGCGGAAGATC |
| CCND-1-R | GAAGACCTCCTCCTCGCAC |
| CTNNB1-F | CTTTGCGTGAGCAGGGTGCCA |
| CTNNB1-R | CCCCCTCCACAAATTGCTGCTG |

### 2.7 Cytotoxicity assay

Cell viability was assessed using 3-(4,5-dimethylthiazol-2-yl)-2,5-diphenyltetrazolium bromide (MTT) assay after 72 h treatment. The treatment lasted 72 h, because this treatment time point resulted in a better outcome (S1 Table). Briefly, cells were incubated with 1[peptide]:1[molecule] combination of the designed dimer peptide and small molecule at different concentrations (50:50 μM, 100:100 μM, 150:150 μM, and 200:200 μM) for 72 h. Then, a solution of MTT in PBS (1 mg/ml) was added to each well to a final concentration of 0.5 μg/μL. After 4 h, incubation at 37 ˚C, precipitated formazan was solubilized with an equal volume of propanol and the absorbance determined at 570 nm. All assays were performed in triplicate.

### 2.8 Quantitative real-time PCR

Total RNA of cell lines was isolated by Tripure (Roche Applied Sciences; Cat No. 11667157001) according to the previously mentioned protocols [26]. Subsequently, cDNA was synthesized by 2μg of RNA, 2U of Mu-MLV (Thermo Scientific, Waltham, Massachusetts, USA; Cat No. EP0441), 10 nmol/l of oligo dT as the primer, and 1U of Ribolock (Thermo Scientific; EO0381). The expression level of CTNNB1 and CCND1 was quantitatively examined by a standard SYBR Green real-time PCR using GAPDH (Humdiagnostics Company, Iran) as the endogenous control. The sequences of designed primers are provided in Table 2. The final reaction was performed in 25μl of reaction mixture containing 12.5μl of SYBR Green Master mix (Takara, Shiga, Japan; Cat No. 4309165), 5pmol of forward and reverse primers, and 2μl of cDNA. The mRNA levels were studied in treated cell lines in comparison with the untreated cells.

## 3. Results and discussions

### 3.1 Oncogenic mutations affect the interaction mode of K-Ras with GTP/GDP

The catalytic domain of K-Ras proto-oncogenes is composed of six β-strands (β1–β6) and six α-helices (α1–α6) flanked by several loops. It has been previously reported that the binding interface of GTP and other effector proteins, including Raf, PI3K, RalGDS, and GAP comprise of three distinct regions called P-loop (between β1 and α1), switch I (between α1 and β2), and switch II (between β3 and α2) which are located in regions 10–17, 32–38, and 59–67 of the protein, respectively (Fig 1A). These regions form a closed conformation upon binding to GTP and are opened after GTP hydrolysis (Fig 1B). In the wild-type K-Ras, residues G13, Y32 and Q61 participate in H-bonding interactions with GTP and initiate the binding and subsequent hydrolysis of this nucleotide. Moreover, interactions between T35 and G60 with the γ-phosphate of GTP stabilize the 'active' state of the nucleotide-protein complex [27]. However, analysis of mutant types of K-Ras deciphered that only G13 and T35 can form H-bonding interactions with GDP suggesting their pivotal role in GTP → GDP hydrolysis even after oncogenic mutations of K-Ras.

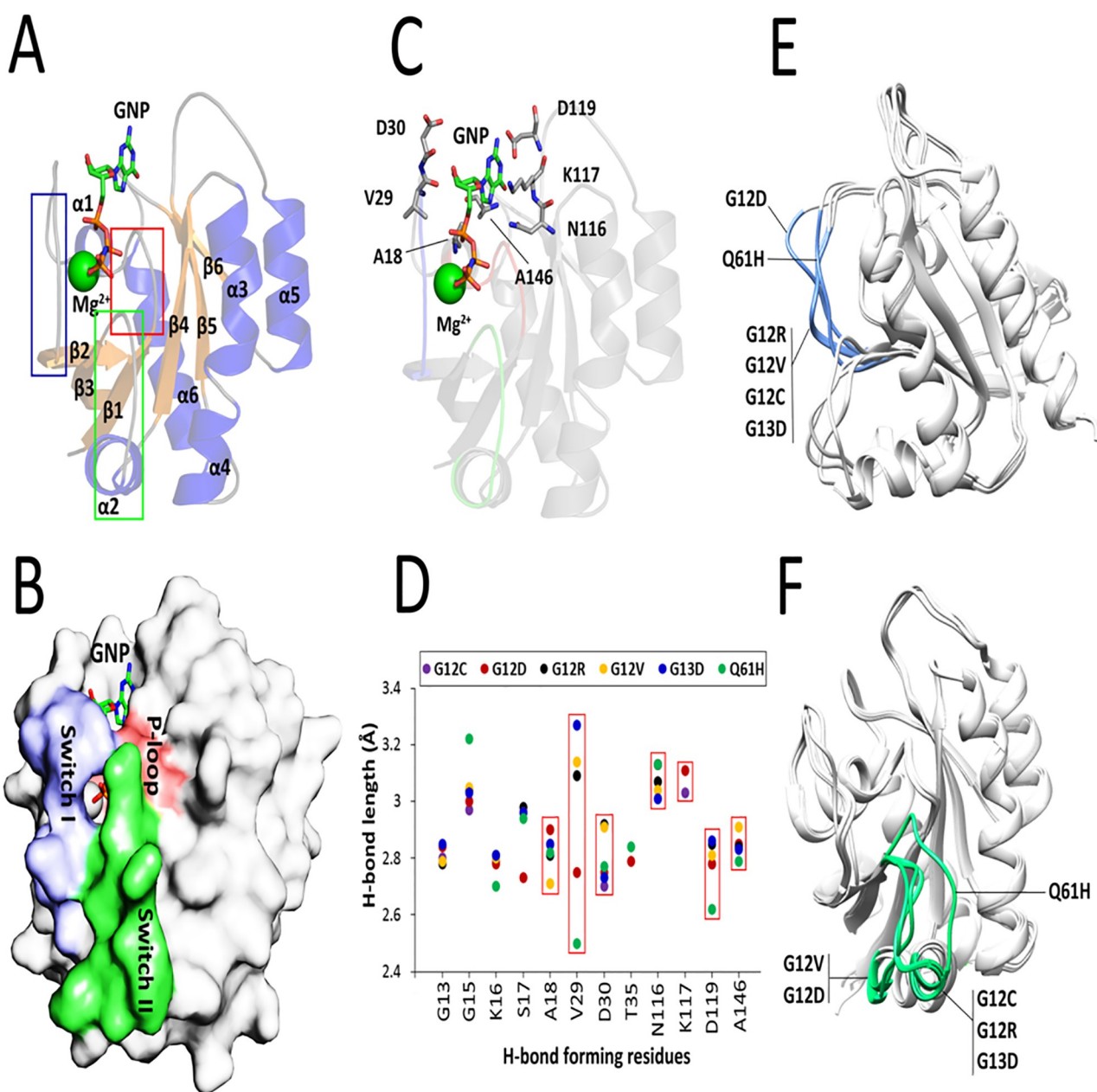

**Fig 1. Assessment of nucleotide-binding site of K-Ras bound to GTP/GDP.** (**A**) Cartoon representation of K-Ras structure bound to GNP (PDB ID: 2PMX). The β-strands, α-helices, and coils are shown in orange, blue, and gray colors, respectively. The P-loop, switch I, and switch II domains are indicated in red, blue, and green boxes, respectively. (**B**) Surface representation of GTP-bound K-Ras along with its nucleotide-binding sites. (**C**) Arrangements of the 7-residue binding interface in the GTP-bound K-Ras. (**D**) Calculation of H-bond distances between the atoms of 12 common nucleotide-binding residues and GTP/GDP in all studied K-Ras mutants. The H-bond length of 7 residues located outside of the previously characterized GTP/GDP-binding site is shown in red boxes. (**E**) Effects of the oncogenic mutations on switch I (blue) and (**F**) switch II (green) domains of K-Ras mutants.

Prediction of H-bonding interactions between mutant types of K-Ras and GTP/GDP revealed that 12 residues of K-Ras including G13, G15, K16, S17, A18, V29, D30, T35, N116, K117, D119, and A146 involved in H-bonding interactions of the protein with GTP/GDP are common between at least two K-Ras mutants. Therefore, these residues presumably stabilize

the complex of GTP/GDP and the mutant-type proteins, and thereby contribute in GTP-to-GDP switches. Interestingly, 7 of these residues including A18, V29, D30, N116, K117, D119, and A146 are located outside the previously reported binding sites of K-Ras implying a novel activity of these residues upon K-Ras oncogenic mutations (Fig 1C). Except for that of N116 and K117, short H-bond distances between the atoms of these novel active residues and GTP/GDP can arguably explain their crucial role in the binding of K-Ras mutants to its effectors as well as providing a platform for hydrolysis of the nucleotide. The V29 showed a meaningful variation in h-bond length among common GTP/GDP-binding residues (Fig 1D). Accordingly, interactions of the small molecules with the aforementioned 12-residue interface were considered as an important criterion in the selection of the final inhibitory small molecules to specifically target the mutant K-Ras proteins.

Although Lu and colleagues have found that the oncogenic mutations result in a substantial conformational change in the nucleotide-binding site of K-Ras4B [27], there is a need to know which mutations have similar effects on the conformation of the nucleotide-binding site of K-Ras. To this end, a structural alignment was carried out between six K-Ras mutants. Intriguingly, only switch I and switch II domains undergo considerable conformational changes upon K-Ras mutations. The G12D and Q61H mutations cause a notably different change in the conformation of switch I. However, the effect of G12R, G12V, G12C, and G13D on the conformation of switch I were approximately similar (Fig 1E). On the other hand, G12D and G12V induced a movement of the switch II which lead to the displacement of α2 in the K-Ras structure. Moreover, Q61H mutation afforded a large shift in switch II domain from 'down' to 'up' conformations. The switch II domain was observed to have approximately the same fold after G12C, G12R, and G13D mutations (Fig 1F). Taken together, G12C, G12R, and G13D mutations have almost a similar effect on the conformation of both switch I and switch II. Furthermore, G12D, G12V, and Q61H mutations induced a considerable change in the switch I and switch II conformation demonstrating their important role in the K-Ras hyperactivity.

After removing the unwanted ions and molecules, the crystal structure of six studied K-Ras mutant-types bound to GTP/GDP were further interpreted using the PDBsum web tool. As G12, G13, and Q61 are located close to the GTP/GDP-binding interface, a mutation in these amino acids can presumably affect the interaction of K-Ras with GTP/GDP. Surface representation of GTP/GDP-bound K-Ras indicated that G12D and G13D mutations are highly capable of shifting the closed conformation of the nucleotide-binding site to the open state facilitating dissociation of GDP. However, the GTP-binding interface of GTP/GDP-bound K-Ras was maintained in closed conformation after G12C, G12R, G12V, and Q61H mutations (S2 Fig). Indeed, in these K-Ras mutants, residues 12 and 13 contribute in the formation of a 'bridge' that delays the nucleotide dissociation when it is hydrolyzed. This is consistent with this report that K-Ras$^{G12D/V/R}$ (94% in pancreatic ductal adenocarcinoma, 76% in CRC, and 91% in lung adenocarcinoma) and K-Ras$^{G13D}$ (1% in pancreatic ductal adenocarcinoma, 20% in CRC, and 6% in lung adenocarcinoma) mutations or, in general, the substitution of glycine with aspartate are classified as the most frequent mutations in Ras-related human cancers [28]. It should be noted that the GTP- and GDP-bound conformations of K-Ras are associated with the 'active' and 'inactive' state of the protein. Consequently, these data suggest that the inactive-to-active conversion of K-Ras$^{G12D}$ and K-Ras$^{G13D}$ takes relatively less time than other studied mutant types. A recent study showed that insertion of aspartate leads to localized flexibility at the insertion site by affecting the electron density [29]. This increased flexibility can explain why the nucleotide-binding interface of K-Ras is readily opened upon substitution of glycine with aspartate at positions 12 or 13. Another reason can be the electrostatic repulsion between D12/D13 and Y32, a necessary residue for maintaining the close state of the K-Ras nucleotide-binding site.

## 3.2 Calculation of the electrostatic surface potential revealed that GTP/GDP is bound to a cavity with positive surface potential

Due to the general assertion that protein-protein interfaces show a conserved electrostatic, hydrophobic, and shape complementarity [30], we calculated the electrostatic surface potential of GTP-bound K-Ras^G12D focusing on the nucleotide-binding site and the oncogenic mutation sites. Results uncovered that GTP is bound to a cavity on the K-Ras surface with a high positive charge density (Fig 2A). Consistent with this, GTP (formal charge = -1) and GDP (formal charge = -2) have a negative surface potential that results in a stable interaction with K-Ras. In all studied K-Ras mutants, residues G15, K16, S17, and A18 were capable of forming H-bonding interactions with the negatively-charged atoms of GTP/GDP (S3 Fig). Analysis of the polar contacts between GDP and K-Ras^G12D revealed that this nucleotide forms a high number of polar contacts with K-Ras implying the electrostatically guided interactions between the nucleotide and K-Ras mutant (Fig 2B).

In the wild-type protein, GTP/GDP is apparently arrested in the nucleotide-binding cavity of K-Ras through direct interactions of G12/G13 with Y32 (Fig 2C). This closed conformation results in a delayed GTP-to-GDP switch. Consistent with this, it has been previously established that the majority of Ras superfamily members have very slow hydrolysis of GTP to GDP [31]. Like glutamate, aspartate and cysteine, tyrosine has an ionizable negative side-chain that affords the potential both to donate and to accept hydrogen bonds [32]. Therefore, we hypothesized that insertion of an aspartate residue at positions 12 or 13 may result in repulsion with Y32 facilitating the open structure of K-Ras's nucleotide-binding site. As expected, a considerable distance between D12-Y32 (~11 Å) and D13-Y32 (~7.6 Å) was observed that supports this hypothesis (S4 Fig). Collectively, insertion of D12/D13 residues with the negatively-charged side chain (instead of G12/G13 with the neutral side chain) might play a considerable role in the structural modification of the nucleotide-binding site of K-Ras^G12/13D through interaction with the residues like Y32, rather than direct interaction with the nucleotides. This is in agreement with previous observations [27]. Thus, during the screening of small molecules, we considered the H-bonding interactions of D12/D13 with the analog molecule as a crucial criterion to disrupt the impact of these residues on the conformation of K-Ras nucleotide-binding site.

## 3.3 The oncogenic mutations site and functional domains of mutant-type K-Ras were determined for targeting

Due to the induction of open conformation in the nucleotide-binding site of K-Ras^G12D and K-Ras^G13D mutants, these mutant types of K-Ras were chosen for subsequent peptide- and molecule-based virtual screening. We aimed to occupy the nucleotide binding site of K-Ras^G12D and K-Ras^G13D focusing on the core GTP/GDP-binding residues that were found to be common between at least two oncogenic K-Ras mutants (Fig 1D). In parallel, the mutation sites of G12, G13, and Q61, as well as Raf-binding site of K-Ras, were considered for direct peptide-based targeting towards blockade of K-Ras downstream pathways. Indeed, though there is a lack of sufficient information about the distinct mechanisms of oncogenic mutations in the induction of K-Ras hyperactivity, occupying the functional groups of substituted residues can be a useful strategy for modulating the function of oncogenic K-Ras. More importantly, 97–99% of K-Ras mutations affect G12, G13, and Q61 residues [33]. Overall, three distinct sites of K-Ras^G12D and K-Ras^G13D were considered for simultaneous peptide- and molecule-based targeting: 1) the positively-charged GTP/GDP-binding site of the protein, partly overlapped with P-loop; 2) a negatively-charged region, called mutation site, which is overlapped with P-loop and switch II domains (residues 12, 13, and 57–63); and 3) a negatively-charged interface of K-Ras, partly overlapped with the switch I (residues 36–41), which is

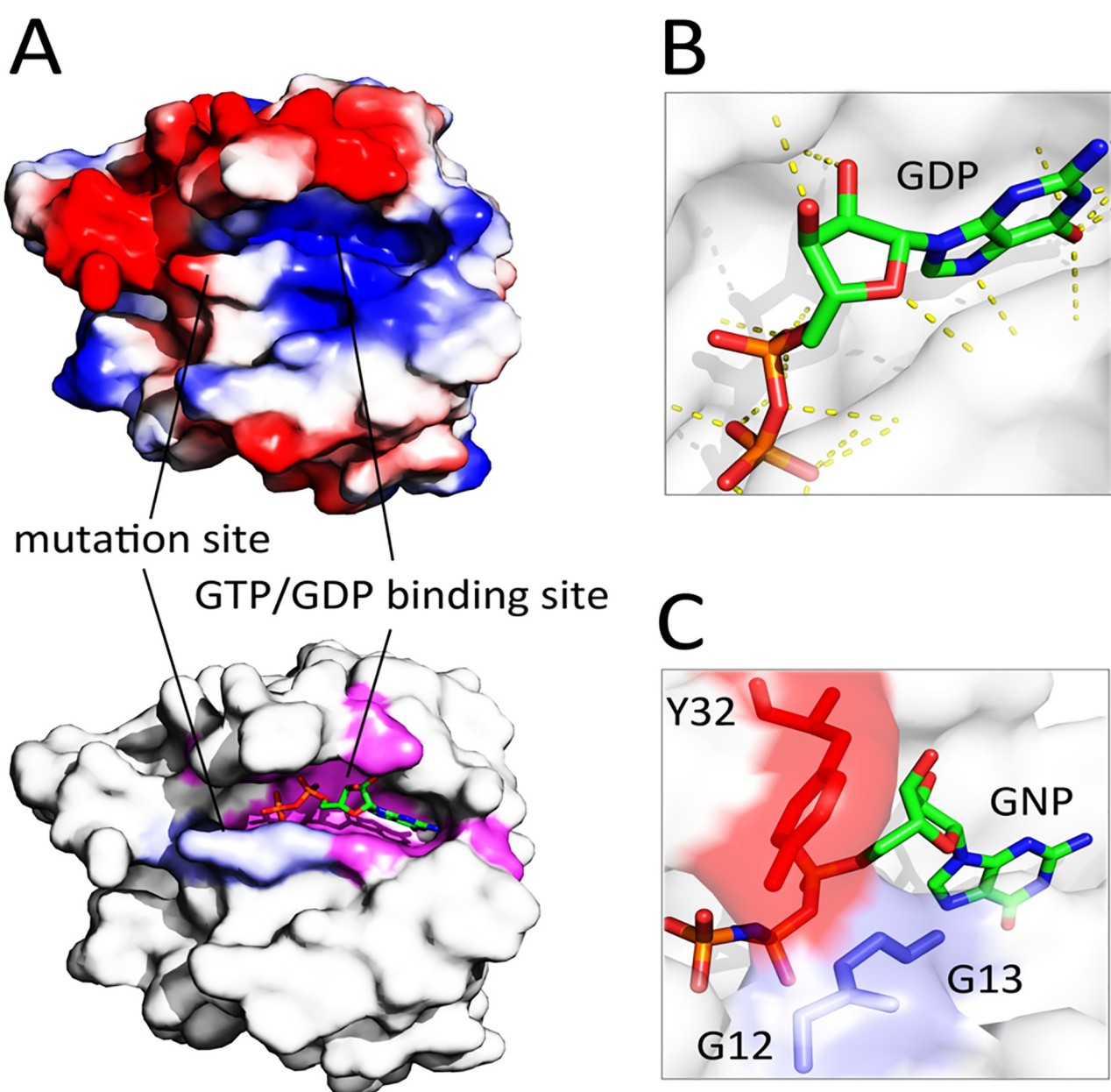

**Fig 2. Evaluation of surface charge pattern of the K-Ras$^{G12D}$.** (**A**) Calculation of the electrostatic surface potential of wild-type K-Ras. The mutation sites 12, 13, and 61 (blue in the figure) and GTP/GDP binding sites (magenta in the below figure) of the protein are shown with solid lines. The blue, red, and gray colors in the above figure refer to the positively-charged, negatively-charged, and hydrophobic regions, respectively (**B**) Polar contacts (yellow dotted lines) between GDP and K-Ras$^{G12D}$. (**C**) A close-up view of the closed conformation of the wild-type GTP-bound K-Ras through direct interaction of G12/13 with Y32.

predominantly involved in the interaction and recruitment of Raf to the plasma membrane [34] (S5 Fig). In parallel, the final selected peptides and small molecule were repeatedly checked for their binding affinity to other studied K-Ras mutants (K-Ras$^{G12C}$, K-Ras$^{G12R}$, K-Ras$^{G12V}$, and K-Ras$^{Q61H}$). The structurally similar hits for GTP/GDP were filtered and used in the docking studies of the first site. Whereas, the positively-charged anti-cancer peptides obtained from the CancerPPD database were optimized for binding affinity and blockade of the mutation and Raf-binding sites.

### 3.4 The library of structurally similar to GNP hits filtered based on desirable absorption, permeability, and oral bioavailability scores

After a similarity screening using the ZINC database, a library including 7305 hits was resulted, which all hits had an approximately >50% same surface and electrostatic properties compared to GNP. Subsequently, the library underwent further filtrations based on the Lipinski, Veber, and Egan rules to omit the compounds with poor absorption, permeability, and oral bioavailability properties. Based on the rule of thumb according to Lipinski et al., a compound with molecular weight (MW) $\leq$ 500, H-bonds donors (HBD) $\leq$ 5, H-bonds acceptors (HBA) $\leq$ 10, and the logarithm of the partition (logP) $\leq$ 5 shows a good absorption or permeability [35]. Whereas, the parameters of the Veber rule (rotatable bonds (RotableB) $\leq$ 10, topological polar surface area (tPSA)$\leq$140 Å, and HBD+HBA$\leq$12), and Egan rule ($0 \geq$ tPSA$\leq$132 and $-1 \geq$ logP $\leq$ 6) for predicting the oral bioavailability are a bit different [36,37]. Based on the values of MW, logP, tPSA, HBA, HBD, and RotableB, 4399 small molecules were rejected from the library (S6 Fig). Moreover, 2432 small molecules were recognized as duplicate hits and filtered from the library. The final library was composed of 474 hits and considered as input for the computational docking studies.

### 3.5 Screening the peptide and small molecule libraries for binding to the corresponding sites

Considering several physicochemical properties as well as the number of assays and cell lines used for validation of anti-cancer activity, a library consisting of 19 peptides was constructed for K-Ras-binding screening. These peptides had a length of 5–16 residues and a molecular weight of 720.9–1952.35 g mol$^{-1}$. Table 1 shows different measured parameters of these peptides.

In order to select the candidate peptides, we screened the peptide library based on the binding affinity and interaction mode of peptides with K-Ras. To this end, molecular docking simulation is a reliable and rapid approach for the identification of peptides with a higher number of hydrogen bonds or other atom contacts with K-Ras [38]. First, we validated the docking method by a previously established structure of K-Ras in complex with an inhibitory peptide (ID: 5XCO) [39]. We re-generated the Kras-peptide complex by molecular docking and compared the results with the crystallography structure of this protein-peptide complex. There was a notable concordance between the docking model and crystallography structure in terms of binding interface and main residues involved in the H-bonding interactions (Fig 3).

Computational docking of the final 19 peptides with K-Ras$^{G12D}$ and K-Ras$^{G13D}$ demonstrated the energetically top-scored peptides for targeting Raf-binding site and mutation site. Among the studied peptides, the LfcinB peptide demonstrated the highest binding energy to the mutation site of K-Ras$^{G12D}$ ($\Delta G_{int}$ = -12.1 kcal mol$^{-1}$, $K_d$ = 1.4e-09). Intriguingly, the binding affinity of this peptide to the mutation site of K-Ras$^{G13D}$ was also notable ($\Delta G_{int}$ = -10.7 kcal mol$^{-1}$, $K_d$ = 7.9e-08). On other hand, the highest binding affinity to the Raf-binding site of K-Ras$^{G12D}$ ($\Delta G_{int}$ = -11.2 kcal mol$^{-1}$, $K_d$ = 6.0e-09) and K-Ras$^{G13D}$ ($\Delta G_{int}$ = -9.9 kcal mol$^{-1}$, $K_d$ = 6.0e-08) was observed in Retro-Kras complex. These peptides had also a considerable binding affinity to other mutant types of K-Ras protein (S2 Table). The results of docking of all peptides with K-Ras$^{G12D}$ and K-Ras$^{G13D}$ are shown in Table 3.

Both Retro and LfcinB were subjected to the affinity maturation process (see section 2.3). Following the affinity maturation steps, a flexible peptide-protein docking was performed between the selected peptides (Retro and LfcinB) and K-Ras$^{G12D/G13D}$ to evaluate the stability of peptides at the binding interfaces of K-Ras mutants. Among top-ranked mutant peptides based on the binding affinity calculations, the peptides with arginine-rich sites in N and/or C

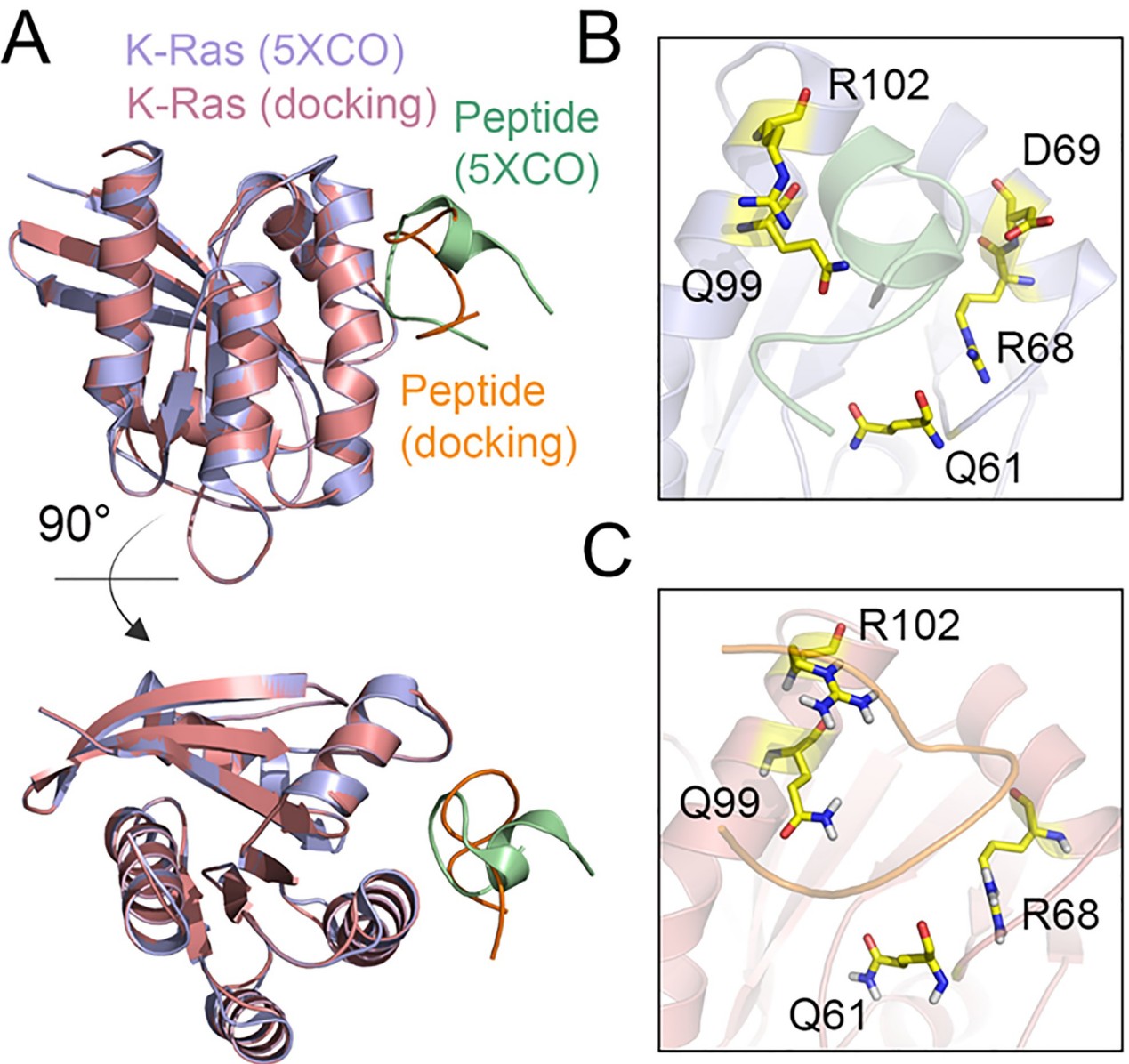

**Fig 3. Validation of molecular docking method.** (**A**) Structural alignment of the K-Ras$^{G12D}$-peptide complex determined by X-ray crystallography (ID: 5XCO) and the docking model. Close-up view of the binding interface between K-Ras$^{G12D}$ and inhibitory peptide in the (**B**) crystallography and (**C**) docking studies. The residues of K-Ras$^{G12D}$ forming H-bond interactions with the peptide are shown in yellow.

terminals were considered with high priority. This priority was due to the high cell-penetrating capability of arginine-rich peptides [40]. According to these criteria, we inserted two arginine residues in the Retro (G3→R3, D13→R13) and LfcinB (C3→R3, M10→R10) sequences (S3 Table). Interestingly, both mutated Retro and LfcinB peptides demonstrated no displacement on the protein binding interfaces implying a stable interaction between the peptides and K-Ras target domains (Fig 4A–4D).

Computational docking of K-Ras$^{G12D}$ with the hits library resulted in the identification of several compounds that could strongly occupy the nucleotide-binding site of K-Ras protein. Among ten top-ranked protein-ligand complexes based on the binding energies (S4 Table), a

**Table 3. Results of docking between peptides and K-Ras$^{G12D}$ and K-Ras$^{G13D}$.** Values of $\Delta G_{int}$ and $K_d$ are presented based on kcal mol$^{-1}$ and molar, respectively. Peptides with highest binding affinity to corresponding interfaces of the proteins are shown in bold.

| Peptide name | K-Ras$^{G12D}$ | | | | K-Ras$^{G13D}$ | | | |
| --- | --- | --- | --- | --- | --- | --- | --- | --- |
| | Mutation site | | Raf-binding site | | Mutation site | | Raf-binding site | |
| | $\Delta G_{int}$ | $K_d$ | $\Delta G_{int}$ | $K_d$ | $\Delta G_{int}$ | $K_d$ | $\Delta G_{int}$ | $K_d$ |
| FLAK93 | -8.0 | 1.4e-06 | -8.1 | 1.1e-06 | -8.5 | 5.8e-07 | -7.9 | 1.5e-06 |
| FLAK50T1 | -9.4 | 1.2e-07 | -8.6 | 1.1e-06 | -8.8 | 3.6e-7 | -9.3 | 1.6e-07 |
| FLAK50T6 | -9.6 | 9.3e-08 | -10.3 | 1.6e-07 | -8.7 | 4.1e-07 | -8.0 | 1.3e-06 |
| FLAK94 | -8.4 | 7.1e-07 | -8.3 | 6.3e-05 | -8.2 | 9.3e-07 | -8.5 | 5.4e-07 |
| C-10 | -10.2 | 3.2e-08 | -8.4 | 7.1e-07 | -10.1 | 4.1e-08 | -9.8 | 5.6e-08 |
| Tat | -7.9 | 1.7e-06 | -9.2 | 1.7e-07 | -9.8 | 6.3e-08 | -8.3 | 8.2e-07 |
| FLAK50 Z1 | -9.2 | 1.7e-07 | -7.5 | 3.4e-06 | -11.1 | 6.8e-09 | -9.7 | 7.8e-08 |
| LfcinB | **-12.1** | **1.4e-09** | **-10.3** | **1.5e-07** | **-10.7** | **7.9e-08** | **-9.8** | **3.5e-07** |
| Temporin-La | -9.0 | 2.6e-07 | -7.5 | 3.0e-06 | -10.2 | 3.3e-08 | -9.0 | 2.5e-07 |
| LL-37(17–29) | -9.2 | 1.7e-07 | -8.2 | 9.8e-07 | -10.2 | 3.4e-08 | -8.9 | 3.2e-07 |
| GA-K4 | -9.5 | 1.0e-07 | -9.0 | 2.4e-07 | -11.1 | 7.2e-09 | -8.8 | 3.4e-07 |
| Cationic Amphiphilic | -8.2 | 1.0e-06 | -7.3 | 4.8e-06 | -9.4 | 1.3e-07 | -9.4 | 1.3e-07 |
| FLAK-120G | -9.2 | 1.8e-07 | -7.2 | 5.0e-06 | -8.4 | 6.7e-07 | -9.0 | 2.5e-07 |
| Halictine 1 | -8.6 | 4.6e-07 | -9.3 | 4.1e-08 | -8.3 | 8.6e-07 | -9.2 | 1.7e-07 |
| (HHPHG)2 | -10.6 | 1.7e-08 | -10.1 | 3.1e-09 | -10.6 | 1.6e-08 | -9.3 | 1.5e-07 |
| 17 | -9.4 | 1.2e-07 | -8.2 | 9.1e-07 | -8.9 | 3.1e-07 | -8.1 | 1.2e-06 |
| CAME-15 | -9.6 | 8.5e-08 | -7.9 | 1.5e-06 | -9.4 | 1.4e-07 | -8.7 | 4.1e-07 |
| RGD-La | -9.4 | 1.3e-07 | -7.0 | 7.0e-06 | -9.5 | 1.0e-07 | -8.8 | 4.2e-07 |
| Retro | **-10.9** | **1.0e-08** | **-11.2** | **6.0e-09** | **-10.5** | **1.9e-08** | **-9.9** | **6.0e-08** |

small molecule with the accession number Zinc12502230 showed more hydrogen bonds and better interactions with targeted residues compared to other small molecules. The binding site of Zinc12502230 has a notable overlap with the tri-phosphate tail of GNP (Fig 4E). Visualization of interactions between Zinc12502230 and K-Ras$^{G12D}$ indicated that this compound forms eight hydrogen bonds with residues D12, G15, K16, S17, and T35 (Fig 4F). Moreover, the binding energies of Zinc12502230-Kras$^{G13D}$, Zinc12502230-Kras$^{G12C}$, Zinc12502230-Kras$^{G12R}$, Zinc12502230-Kras$^{G12V}$, and Zinc12502230-Kras$^{Q61H}$ were calculated as -10.1 kcal mol$^{-1}$, -8.8 kcal mol$^{-1}$, -9.3 kcal mol$^{-1}$, -11.0 kcal mol$^{-1}$, and -9.8 kcal mol$^{-1}$, respectively. This confirms the high binding affinity of the small molecule Zinc12502230 to other K-Ras mutants.

According to the deep analysis of docking results, a dimer peptide encompassing optimized Retro and LfcinB peptides was considered for simultaneously targeting the switch I and switch II of K-Ras mutants. The GGGGS peptide was used as a flexible linker to inhibit conformational interference of each peptide with another one. A conventional procedure for improving the cellular uptake of peptides is the fusion of that peptides to arginine-rich cell-penetrating peptides (CPPs) [41]. Therefore, during affinity maturation steps, in addition to the peptide-Kras binding affinity, we also aimed to enhance the sequence similarity between the Retro and LfcinB peptides and previously established CPPs through the insertion of arginine residues to enhance their intracellular delivery. Recently, arginine-rich peptides have also demonstrated stable interaction with mutant-type K-Ras resulting in the successful blockade of the K-Ras Raf-binding site [39]. However, only a few substitutions were performed to preserve the original properties of each peptide (S3 Table). Similar to our mutated LfcinB peptide, the peptide in the 5XCO crystallography file has Arginine-rich sequences in both N and C terminals [39].

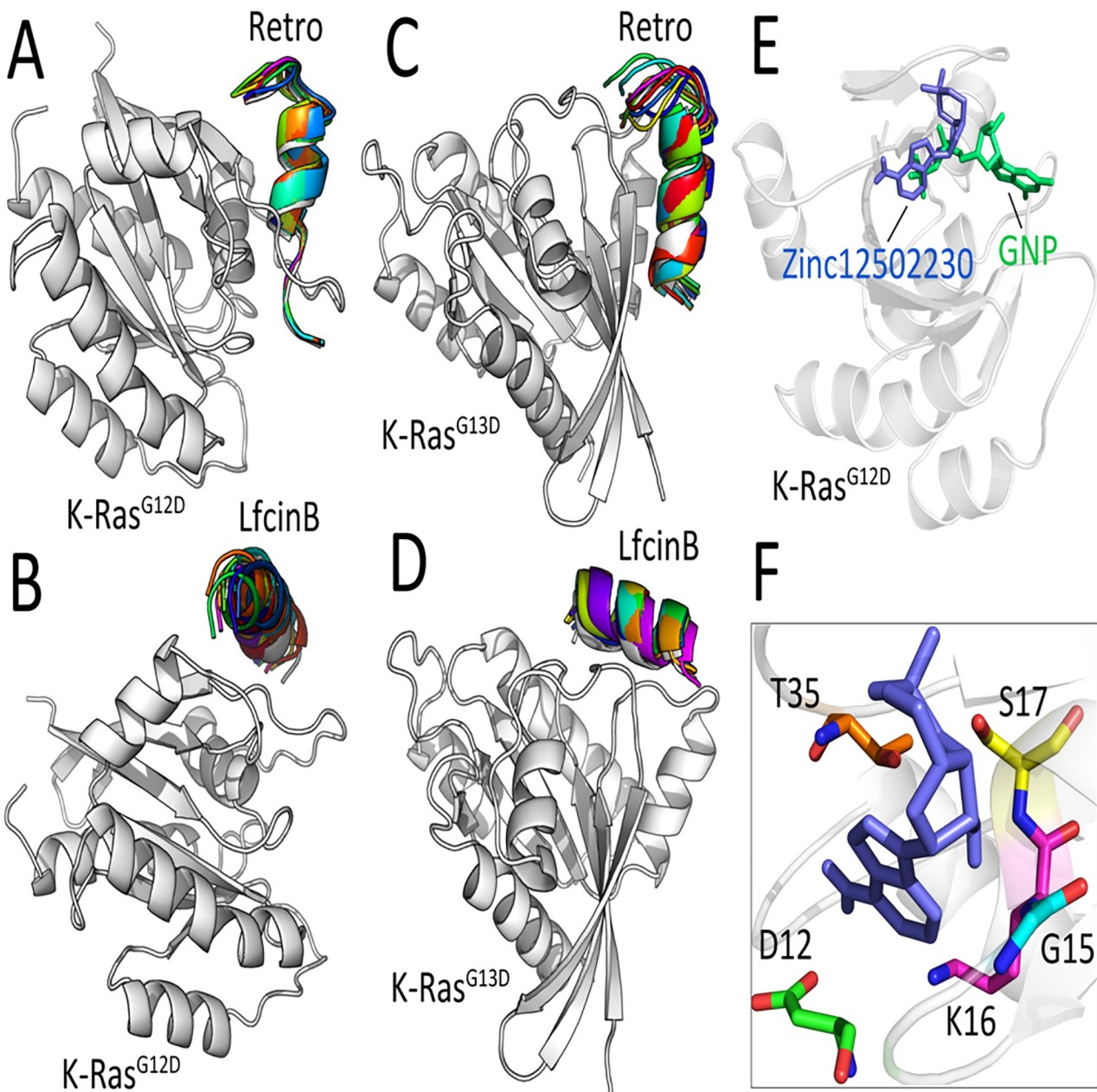

**Fig 4. Interaction mode of peptides/molecule with K-Ras^G12D and K-Ras^G13D.** Superimposition of top-ranked peptide-protein docked poses resulted from flexible docking of (**A**) the modified Retro-Kras^G12D, (**B**) the modified LfcinB- Kras^G12D, (**C**) the modified Retro-Kras^G13D, and (**D**) the modified LfcinB-Kras^G13D. (**E**) Superimposition of GNP-bound (green) and Zinc12502230-bound (blue) K-Ras^G12D. (**F**) Close-up view of H-bonding interactions between Zinc12502230 (blue) and five residues of K-Ras^G12D colored in different colors.

This is actually a validation of our designed peptides, since during affinity maturation changes, we tried to increase the number of R/K residues in both N and C terminal of the peptides to enhance their cell-penetrating properties. Similar to our designed peptide, the peptide in the 5XCO binds to the K-Ras near to the switch domains. The final designed dimer peptide was predicted to be cell-penetrating based on the support vector machine (SVM)-based models, which utilizes a validated dataset and various features such as amino acid composition,

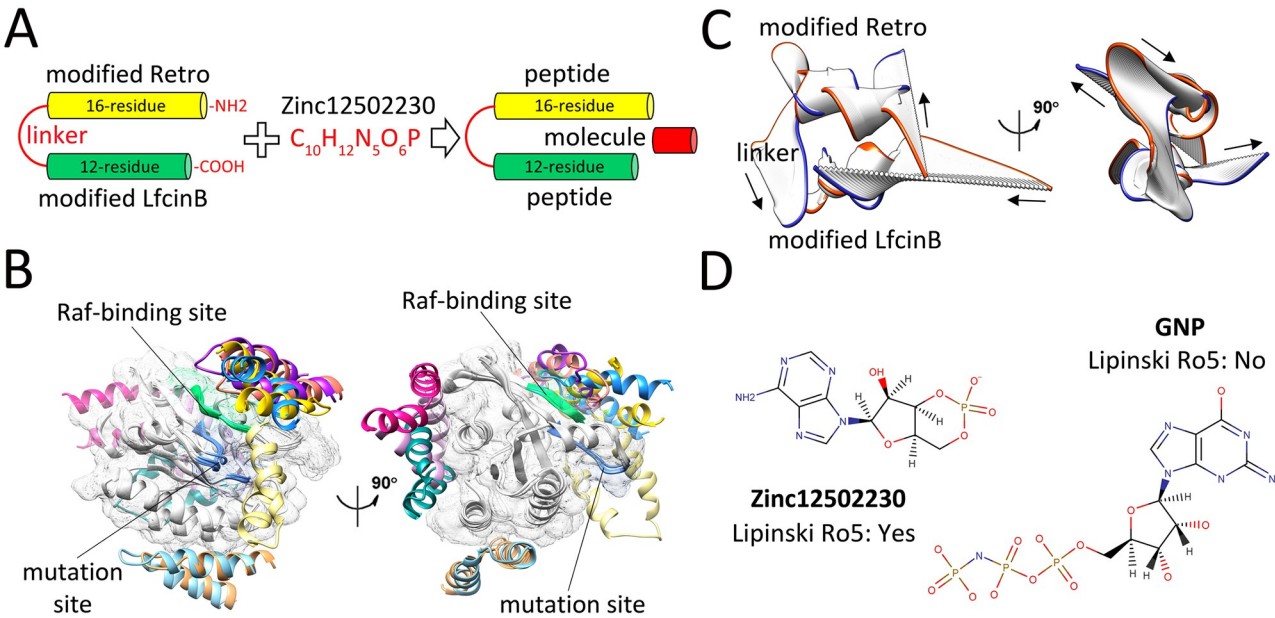

**Fig 5. Binding energy and structural evaluations of final selected peptides and molecule.** (**A**) Schematic representation of dimer peptide construction and combination of this peptide with Zinc12502230. (**B**) Ten top-ranked peptide-protein complexes obtained from flexible docking of the modified Retro-LfcinB peptide (each pose in a different color) with K-Ras$^{G12D}$ (gray). The Raf-binding and mutation sites are shown in green and blue colors, respectively. (**C**) The dominant mode of motion resulted from 50 frames of Retro-LfcinB MD simulations. The orange and blue colors represent the first and last frames, respectively. Arrows show the direction of motions. (**D**) Molecular representation of the small molecule Zinc12502230 (left) and GNP (right).

dipeptide composition, the binary profile of patterns, and physicochemical properties of predictions [42].

The modified Retro-LfcinB dimer peptide was subsequently combined with small molecule Zinc12502230 in different concentrations to block oncogenic K-Ras (Fig 5A). Prior to this step, a blind flexible docking was carried out between the modified Retro-LfcinB dimer peptide and K-Ras$^{G12D}$ to investigate the binding specificity of this dimer peptide to the relevant target sites on the protein surface. Among ten top-ranked complexes, five complexes illustrated a specific interaction of Retro-LfcinB peptide with Raf-binding and mutation sites of K-Ras$^{G12D}$ ($\Delta G_{int} < $ -9 kcal mol$^{-1}$) suggesting that this dimer peptide can specifically occupy the regions of interest on the surface of K-Ras mutants (Fig 5B). More interestingly, three other top-scored peptide-protein complexes (magenta, pink and dark cyan colors in Fig 5B) indicated that the designed dimer peptide could also interact with a C-terminal region (124–170) of K-Ras$^{G12D}$. The C-terminal region of K-Ras (residues 166–180) has been previously characterized as a hypervariable region (HVR) which is responsible for plasma membrane targeting of the protein through direct interaction with some other proteins, such as Calmodulin, a Ca$^{2+}$-binding protein that can bind to and regulate the function of several target proteins eg., phosphodiesterase and protein kinases [43,44]. Consequently, the Retro-LfcinB dimer peptide may also play an inhibitory role in K-Ras plasma membrane translocation through modulating the interaction of this protein with its partners. However, this claim needs to be investigated in more detail. It is worth note that the Retro-LfcinB dimer peptide could strongly bind to other studied K-Ras mutants, especially K-Ras$^{Q61H}$ ($\Delta G_{int} = $ -16.0 kcal mol$^{-1}$, $K_d = $ 6.2e-09) (S7 Fig).

The principal components analysis (PCA) was performed on MD trajectories to estimate the essential motions of Retro-LfcinB dimer peptide. This method is one of the most used

techniques to capture the dominant time-dependent motions in a given macromolecule utilizing MD trajectories [45]. A notable motion was observed in the linker region suggesting that this region is flexible enough to allow free movements of modified Retro and LfcinB on the K-Ras surface (Fig 5C). Due to the high flexibility, the favorable binding of a peptide to its binding site might be served as an anchor for better interaction of other peptides with the K-Ras protein.

Structural comparison of Zinc12502230 and GNP deciphered some similarities in their structures. Indeed, the majority of atoms in Zinc12502230 and GNP lie on the same plane defined by the pentagon and hexagon ring structures. However, in contrast to GNP, Zinc12502230 has acceptable Lipinski's rule of five (Ro5) parameters. (Fig 5D). Intriguingly, such ring structures can also be found in other small molecule inhibitors of K-Ras. For instance, Sun and colleagues found several K-Ras inhibitors that have similar structures to the Zinc12502230. These compounds were highly capable of occupying a hydrophobic pocket of K-Ras located between switch I and switch II and thereby could inhibit Sos-catalyzed K-Ras activation [46].

Both global and site-directed docking for K-Ras–peptide complexes were performed. In the global docking, a specific binding affinity was observed for the peptide to mutant K-Ras (Fig 5B). For site-directed docking, the Flex-pep dock algorithm was used and that analysis also confirmed the specific binding affinity of dimer peptide to the mutation and Raf-binding sites. However, two selected peptides had a high affinity for both mutation and Raf-binding sites. This was actually a good finding in our design since the main goal of designing a dimer peptide was to occupy both target sites, simultaneously. Therefore, it was great to see that two selected peptides have specific affinities to target domains, with higher affinity to one site but also high binding affinity to another site. So that they apparently can block the target sites more efficiently. PCA analysis proved the flexibility of each peptide around the linker sequence indicating that peptides could readily rotate around the linker.

To examine the dynamics and stability of K-Ras protein in the free and complex forms, we analyzed the Root Mean Square Deviation (RMSD) and Root Mean Square Fluctuation (RMSF) by MD trajectories resulting from a short MD simulation (10ns). Although the backbone RMSD of K-Ras in the complex form showed fluctuations at the beginning, the structure became stable at the last nanoseconds of the MD simulation. The variation of the RMSD graph was between 0 to 1.5Å (Fig 6A). To explore the impact of the designed peptide on the dynamics of the side-chain atoms, RMSF plots were also generated. In general, there were fewer fluctuations for the K-Ras residues in the complex form compared to the free protein, especially in

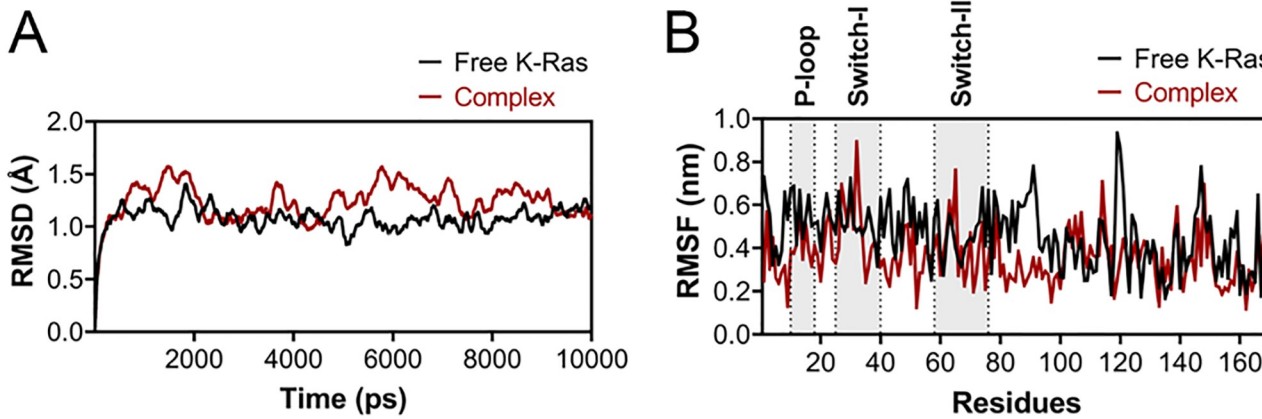

**Fig 6. RMSD and RMSF plots of free K-Ras and Kras-peptide complex.** (**A**) The RMSD and (**B**) RMSF of plots of free (black) and complex (red) K-Ras forms after a 10 ns MD simulation. The position of target domains is shown the B panel.

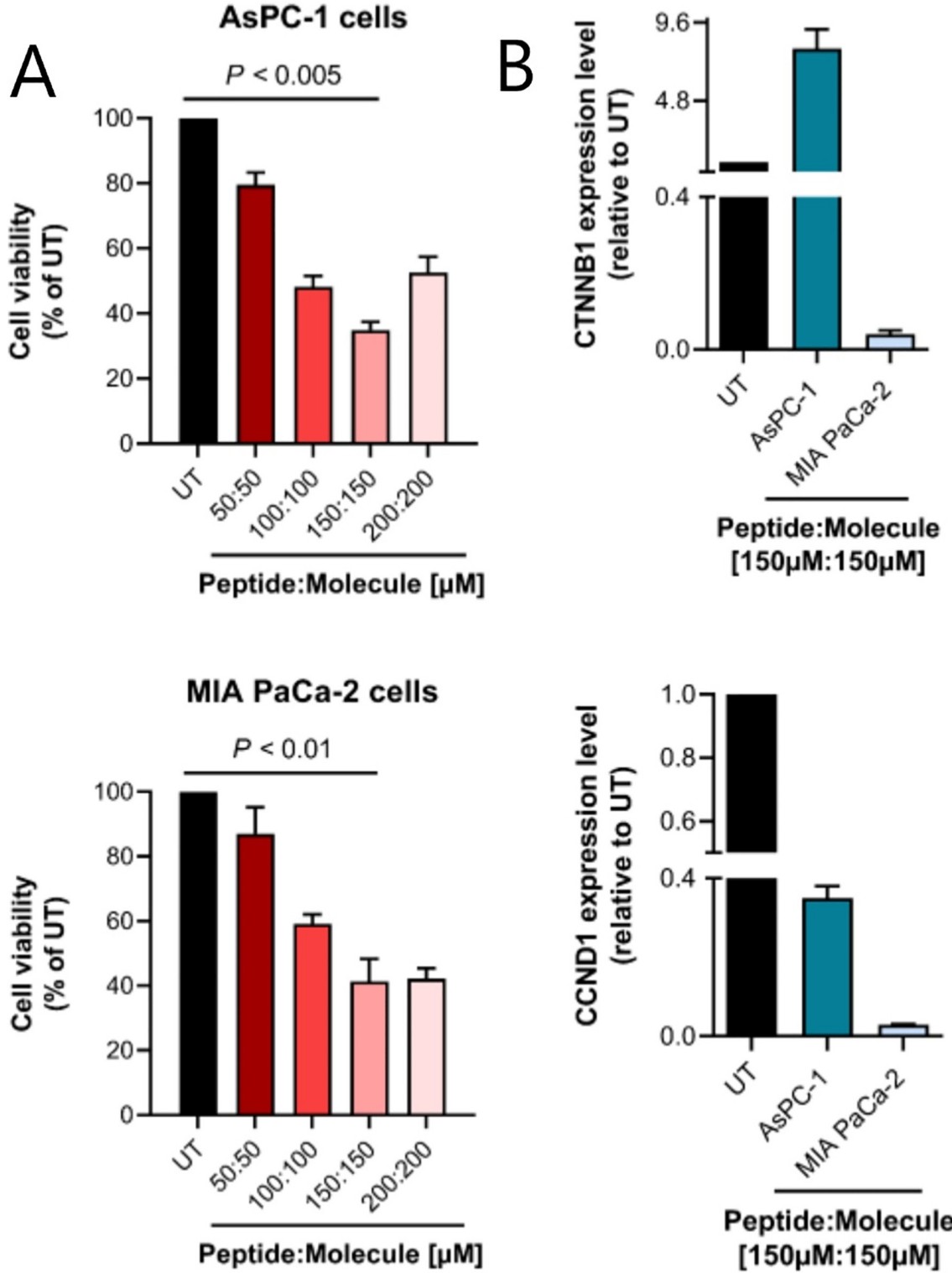

**Fig 7. MTT assay and quantitative real-time PCR results.** (**A**) Assessment of the cell viability of AsPC-1 (up) and MIA PaCa-2 (down) cells after treatment with different concentrations of Retro-LfcinB peptide and Zinc12502230 small molecule. Values were normalized with untreated cells. The highest cytotoxicity of the compounds (at 150µM of peptide: 150µM of the small molecule) was significantly different from untreated control at $P < 0.005$ and $P < 0.01$ (Student's t-test) in AsPC-1 and MIA PaCa-2 cells, respectively. UT: untreated cells. (**B**) Cancer cells expressing K-Ras[G12D] were treated with the combination of 150µM of modified Retro-LfcinB peptide and 150µM of Zinc12502230 small molecule during 72 h, and then mRNA level was analyzed with quantitative real-time PCR for expression of K-Ras down-stream genes including CTNNB1 and CCND1. Results are the mean ± SE of three separate experiments.

the studied domains. This can be attributed to the local stability of K-Ras domains induced by the peptide. Interestingly, the dynamics of residues in the free K-Ras C-terminal was highly overlapped with the complex form (Fig 6B). This is compatible with the interaction mode of the peptide to K-Ras, which is bound to its target domains located in the N-terminal of the protein, but not the C-terminal.

### 3.6 Retro-LfcinB and Zinc12502230 decreased the viability of Kras-positive pancreatic cancerous cells

The inhibitory effect of the designed anti-Kras peptide and small molecule was assessed on the K-ras-positive pancreatic cancer cell lines AsPC-1 and MIA PaCa-2. The results indicated that both cancer cell lines were susceptible to the combination of Retro-LfcinB and Zinc12502230. The combination of the compounds resulted in a dose-dependent decrease in the viability of tested K-ras-positive cancer cells. Particularly, the highest cytotoxicity of the compounds was observed at the combination of 150µM of Retro-LfcinB peptide with 150µM of Zinc12502230 molecule, which resulted in ~65% and ~59% cell death in AsPC-1 and MIA PaCa-2 cell lines, respectively ($P < 0.005$ for AsPC-1 cells and $P < 0.01$ for MIA PaCa-2 cells compared to the untreated group) (Fig 7A). However, especially in co-culture with AsPC-1 cells, a decreased efficacy of Retro-LfcinB and Zinc12502230 was observed at higher concentrations (200µM of the peptide: 200µM of the small molecule). This phenomenon can be attributed to the formation of colloidal aggregates of the peptide and/or small molecule at higher concentrations limiting their efficacy [47].

### 3.7 Retro-LfcinB and Zinc12502230 down-regulated CTNNB1 and CCND1 in Kras-positive cancer cells

Studies have previously established the central role of oncogenic K-Ras in the regulation of transcription factors involved in the Wnt/β-catenin signaling pathway [3]. Moreover, CCND1 is known as a downstream effector of K-Ras which regulates cell division during G1-S transition of the cell cycle [48]. Here, we evaluated the expression level of two important K-Ras downstream regulators, CTNNB1 and CCND1 following blockade of K-Ras in the examined cancer cells. The designed K-Ras inhibitors significantly decreased the mRNA expression of these genes, especially in MIA PaCa-2 cells. Down-regulation of CCND1 was highly remarkable in both K-ras-positive cancer cell lines (Fig 7B). Unexpectedly, however, CTNNB1 was up-regulated in the treated AsPC-1 cells. This can be explained by the 'hub' function of CTNNB1, which can be controlled by multiple regulators.

In conclusion, our study demonstrates that simultaneous targeting of K-Ras oncogenic mutations and functional domains shows encouraging outcome and can be considered as a novel strategy for the treatment of K-ras-related cancers. This finding is remarkable since K-Ras is still known as an 'untargetable' oncogene in cancer and targeting of single domains/ regions of K-Ras via small peptides and molecules has failed to result in significant clinical benefits. However, the main part of this study includes bioinformatics analyses, and therefore, further experimental validations are required.

## Supporting information

**S1 Fig. Calculation of surface electrostatic potential of studied peptides.** The positively-charged, negatively-charged and hydrophobic regions are shown in blue, red and gray colors, respectively.
(TIF)

**S2 Fig. The effects of oncogenic mutations in G12, G13 and Q61 positions on the dynamics of GTP/GDP-binding site of K-Ras.** Surface representation of the structures of GDP-bound (**A**) K-Ras$^{G12C}$, (**B**) K-Ras$^{G12D}$, (**C**) K-Ras$^{G12R}$, (**D**) K-Ras$^{G12V}$, (**E**) K-Ras$^{G13D}$ and (**F**) K-Ras$^{Q61H}$. The mutation sites and GTP/GDP-binding sites are shown in blue and magenta colors, respectively.
(TIF)

**S3 Fig. Representation of interactions between GNP/GDP and K-Ras residues.** The surface electrostatic potential of GNP/GDP are shown in complex with K-Ras residues (sticks). The nucleotide atoms with positively-charged, negatively-charged and hydrophobic properties are shown in blue, red and gray colors, respectively.
(TIF)

**S4 Fig. Calculation of approximate distance between Y32 and D12/13 in K-Ras$^{G12D}$ and K-Ras$^{G13D}$.** Interaction between Y32 and G12/13 in the wild-type K-Ras is also shown.
(TIF)

**S5 Fig. Surface electrostatic potentials of drug targeting sites.** (**Up**) The surface representation of K-Ras$^{G12D}$ and (**down**) K-Ras$^{G13D}$ along with their surface electrostatic potentials. Red, blue and gray colors in the left sides refer to negatively-charged, positively-charged and hydrophobic regions of the proteins, respectively. The Raf-binding site, GTP/GDP-binding site and mutation site are shown in green, pink and blue colors, respectively, in the right side of Fig.
(TIF)

**S6 Fig. The final library of GNP-based hits after several filtering steps.** Distribution of (**A**) MW, (**B**) logP, (**C**) HBD, (**D**) HBA, (**E**) tPSA and (**F**) RotableB after FAFDrugs filtering based on Lipinski, Veber and Egan rules. Threshold of each parameter is shown with a red line.
(TIF)

**S7 Fig. Interaction mode of Retro-LfcinB dimer peptide and K-Ras mutants.** (**A**), (**B**), (**C**) and (**D**) refer to complex of the dimer peptide and K-Ras$^{G12C,}$ K-Ras$^{G12R}$, K-Ras$^{G12V}$ and K-Ras$^{Q61H}$. The Raf-binding and mutation sites of K-Ras are shown in green and blue colors, respectively. The binding energy of each peptide-protein complex are also provided.
(TIF)

**S1 Table. Percentage of AsPC1 cell death in the presence of 100 μM peptide.** Molecule as determined by MTT assay at different culture times.
(DOCX)

**S2 Table. Calculation of binding energy between peptides mutation site/Raf-binding site of K-Ras mutants.**
(DOCX)

**S3 Table. The original and mutated sequence of LfcinB and Retro peptides.**
(DOCX)

**S4 Table. Ten top-ranked compounds in case of their binding affinity to K-Ras$^{G12D}$.**
(DOCX)

## Author Contributions

**Conceptualization:** Mansour Poorebrahim, Ladan Teimoori-Toolabi.

**Data curation:** Mohammad Foad Abazari, Leila Moradi, Hourieh Kalhor, Hassan Askari.

**Formal analysis:** Mansour Poorebrahim, Reza Mahmoudi.

**Funding acquisition:** Ladan Teimoori-Toolabi.

**Investigation:** Mohammad Foad Abazari, Leila Moradi, Behzad Shahbazi.

**Methodology:** Mansour Poorebrahim, Leila Moradi.

**Project administration:** Ladan Teimoori-Toolabi.

**Resources:** Ladan Teimoori-Toolabi.

**Software:** Hourieh Kalhor.

**Supervision:** Ladan Teimoori-Toolabi.

**Validation:** Mansour Poorebrahim.

**Visualization:** Mansour Poorebrahim.

**Writing – original draft:** Mansour Poorebrahim.

**Writing – review & editing:** Ladan Teimoori-Toolabi.

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
