## [Decision Letter · Decision Letter 0]

21 Oct 2021

Dear Dr Teimoori-Toolabi,

Thank you very much for submitting your manuscript "Multi-targeting of K-Ras domains and mutations by inhibitory peptide and small molecules" for consideration at PLOS Computational Biology.

As with all papers reviewed by the journal, your manuscript was reviewed by members of the editorial board and by several independent reviewers. In light of the reviews (below this email), we would like to invite the resubmission of a significantly-revised version that takes into account the reviewers' comments.

We cannot make any decision about publication until we have seen the revised manuscript and your response to the reviewers' comments. Your revised manuscript is also likely to be sent to reviewers for further evaluation.

Sincerely,

Alexander MacKerell

Associate Editor

PLOS Computational Biology

Nir Ben-Tal

Deputy Editor

PLOS Computational Biology

Reviewer's Responses to Questions

**Comments to the Authors:**

Reviewer #1: The manuscript by Mansour et al. “K-Raz targeting by peptide and molecule inhibitors” describes peptides and small molecules development as K-Ras inhibitors using different computational methods along with in-vitro study. The manuscript is well written and described high-quality research using advanced methodologies of molecular modelling and in-vitro assays. I recommend the article for publication in Plos Computational Biology after minor revisions.

Comments to the Authors

1. It will be good to include some types of validation of the docking protocol in the manuscript.

2. The RMSd and RMSf graphs should be included in the manuscript to know about the stability and flexibility of the complex structures.

3. The author should read the recently published article about K-ras by Mehreen et al, 2021, Journal of Biomolecular Structure and Dynamics, vol.38, 5488.

4. Throughout the manuscript small molecules should write instead of molecules

Reviewer #2: In the current work, the authors computationally screened peptide and small molecule databases to find peptide and small molecules targeting Kras. And based on the docking poses of top ranked peptides, they designed a peptide dimer. The inhibitory effect of the designed peptide dimer and molecule was assessed on the K-ras-positive pancreatic cancer cell lines. This work is novel, however, the current result is not fully supported by the presented data. The authors are recommended to consider the following comments to enhance the work.

1.The authors docked 19 peptides for Kras and found two top ranked peptide LfcinB and Retro targeting mutation site and Raf-binding site. While the LfcinB has more favorable docking score for mutation site over the Raf-binding site. The Retro has very similar docking scores for both sites. Only the Raf-binding site binding pose of Retro was considered as shown in Figure 3A and 3C. Did the author analyze the docking poses of Retro for both sites? For the peptide docking, did the author tried the global docking or only the mutation site and Raf-binding site being considered? Since the Kras protein size is quite trackable, MD simulations on the predicted peptide-protein complexes could help to further verify the docking poses.

2. Affinity maturations on the two top ranked peptide were performed in the current work, however, no discussion was made. Figure 3 shows the overlaid docking poses of different designs of the two peptides. What are the mutations being made on the two peptides? How does such mutations help to improve the binding? More discussions are required here.

3. Based on the predicted docking poses of the two top ranked peptides, the authors designed peptide dimer to utilize the binding benefits targeting both the mutation and Raf-binding sites. And they addressed that "we aimed to enhance the sequence similarity between the Retro-LfcinB dimer peptide and previously established CPPs to increase its intracellular delivery". Again, there is zero information about what mutations were conducted to modify the peptide dimer to enhance its profile. More information is required at here. And again, MD simulation on the predicted peptide-protein complex could help to verify the docking results.

4. The author mentioned "Molecular Dynamics (MD) trajectories were generated for both selected peptide-protein and molecule-protein complexes" in the Method section. However, the only discussion related to MD simulations is a PCA analysis on the peptide dimer to prove the flexibility of the linker. Does this done for MD trajectories on the peptide dimer alone or with protein? More analyses are required to fully support the design.

5, There is actually a peptide inhibitor-Kras complex crystal structure available under PDB entry: 5XCOm (acsmedchemlett.7b00128). It is also an arginine rich peptide design. It changes the K-ras confirmation at the switch II. Is there any similarity between this design with the authors? Again, the authors should show their final peptide design. Is there any information can be borrowed from this crystal structure to interpret the current work?

6. Since the cell line inhibitory is not a direct binding assay, it is hard to tell how does the designed peptide dimer and the small molecule work together to bind K-ras. The authors can easily do another set of assay tests to verify the effects brought by the potential joint binding. Comparison between inhibitory effects brought by using the peptide dimer or molecule alone with the current inhibitory results using joint peptide:molecule can further elucidate the design of using both peptide and small molecule for Kras.

**Have the authors made all data and (if applicable) computational code underlying the findings in their manuscript fully available?**

Reviewer #1: Yes

Reviewer #2: None

PLOS authors have the option to publish the peer review history of their article (what does this mean?). If published, this will include your full peer review and any attached files.

Reviewer #1: No

Reviewer #2: No
---

## [Decision Letter · Decision Letter 1]

24 Feb 2022

Dear Dr Teimoori-Toolabi,

We are pleased to inform you that your manuscript 'Multi-targeting of K-Ras domains and mutations by peptide and small molecule inhibitors' has been provisionally accepted for publication in PLOS Computational Biology.

Best regards,

Alexander MacKerell

Associate Editor

PLOS Computational Biology

Nir Ben-Tal

Deputy Editor

PLOS Computational Biology

Reviewer's Responses to Questions

**Comments to the Authors:**

Reviewer #2: The authors addressed most of my comments except for the last one. But based on their funding situation and computational part was much better addressed in this version which is the main aim of the journal, I think it is ready for publication.

**Have the authors made all data and (if applicable) computational code underlying the findings in their manuscript fully available?**

Reviewer #2: None

PLOS authors have the option to publish the peer review history of their article (what does this mean?). If published, this will include your full peer review and any attached files.

Reviewer #2: No

---

## [Editor Report · Acceptance letter]

31 Mar 2022

PCOMPBIOL-D-21-01581R1 

Multi-targeting of K-Ras domains and mutations by peptide and small molecule inhibitors

Dear Dr Teimoori-Toolabi,

I am pleased to inform you that your manuscript has been formally accepted for publication in PLOS Computational Biology. Your manuscript is now with our production department and you will be notified of the publication date in due course.

With kind regards,

Zita Barta
